# Not discussed: Inequalities in narrative text data for suicide deaths in the National Violent Death Reporting System

**Briana Mezuk**[1,2]*, **Viktoryia A. Kalesnikava**[1], **Jenni Kim**[3], **Tomohiro M. Ko**[1,4], **Cassady Collins**[1]

**1** Department of Epidemiology, University of Michigan School of Public Health, Ann Arbor, Michigan, United States of America, **2** Research Center for Group Dynamics, Institute for Social Research, University of Michigan, Ann Arbor, Michigan, United States of America, **3** Department of Biostatistics, University of Michigan School of Public Health, Ann Arbor, Michigan, United States of America, **4** Robert Wood Johnson Medical School, Rutgers University – New Brunswick, New Brunswick, New Jersey, United States of America

* bmezuk@umich.edu

**Data Availability Statement:** The narrative data used in this analysis are available by request from the CDC through their restricted-access data process. Our use of these restricted-access

## Abstract

### Background

The rate of suicide in the US has increased substantially in the past two decades, and new insights are needed to support prevention efforts. The National Violent Death Reporting System (NVDRS), the nation's most comprehensive registry of suicide mortality, has qualitative text narratives that describe salient circumstances of these deaths. These texts have great potential for providing novel insights about suicide risk but may be subject to information bias.

### Objective

To examine the relationship between decedent characteristics and the presence and length of NVDRS text narratives (separately for coroner/medical examiner (C/ME) and law enforcement (LE) reports) among 233,108 suicide and undetermined deaths from 2003–2017.

### Methods

Generalized estimating equations (GEE) logistic and quasi-Poisson modeling was used to examine variation in the narratives (proportion of missing texts and character length of the non-missing texts, respectively) as a function of decedent age, sex, race/ethnicity, education, marital status, military history, and homeless status. Models adjusted for site, year, location of death, and autopsy status.

### Results

The frequency of missing narratives was higher for LE vs. C/ME texts (19.8% vs. 5.2%). Decedent characteristics were not consistently associated with missing text across the two types of narratives (i.e., Black decedents were more likely to be missing the LE narrative but

NVDRS data is governed by a Data Use Agreement (DUA) with the CDC. This DUA legally prohibits us from sharing these data with outside investigators. Any investigator can gain access to these restricted access NVDRS data by contacting nvdrs-rad@cdc.gov and following the procedures outlined here: https://www.cdc.gov/violenceprevention/datasources/nvdrs/dataaccess.html Other NVDRS data are publicly-available: https://www.cdc.gov/violenceprevention/datasources/nvdrs/datapublications.html. Cells with <5 observations have been suppressed in this publication, as required by the NVDRS Data Use Agreement.

**Funding:** This project was supposed by the National Institute of Mental Health (R21-108989, https://www.nimh.nih.gov/) and the American Foundation for Suicide Prevention (DIG-1-110-19, https://afsp.org/), both to B. Mezuk. The funders had no role in the conceptualization, analysis, interpretation, or decision to publish this manuscript.

**Competing interests:** The authors have declared that no competing interests exist.

less likely to be missing the C/ME narrative relative to non-Hispanic whites). Conditional on having a narrative, C/ME were significantly longer than LE (822.44 vs. 780.68 characters). Decedents who were older, male, had less education and some racial/ethnic minority groups had shorter narratives (both C/ME and LE) than younger, female, more educated, and non-Hispanic white decedents.

## Conclusion

Decedent characteristics are significantly related to the presence and length of narrative texts for suicide and undetermined deaths in the NVDRS. Findings can inform future research using these data to identify novel determinants of suicide mortality.

## Introduction

In the US, the rate of suicide has increased by more than a third since 1999 [1], despite ongoing and renewed efforts by governmental and non-governmental stakeholders to support research on developing more effective prevention measures [2–5]. Leaders in the field have argued:

> "By and large, the [suicidal thoughts and behaviors (STB)] risk factor field appears to have conducted essentially the same studies over and over again throughout the last 50 years. In light of this pattern, it is not surprising that predictive ability has remained nearly constant over the last 50 years"

[6].

This critique calls for new conceptual models, data sources and analytic approaches to understanding suicidal behavior, with attention to identifying modifiable determinants over the life course.

The National Violent Death Reporting System (NVDRS) is a state-based mortality registry implemented by the CDC that seeks to link "information about the "who, when, where, and how" from data on violent deaths [suicide, homicide, accidental firearm] and provides insights about "why" they occurred" [7, 8]. It is the most comprehensive surveillance system of the circumstances surrounding suicide mortality in the US, and it has recently been expanded to cover all 50 states [9]. The rationale for this rich data source is to enhance investigations that seek to clarify the circumstances and help discern contributing factors for completed suicide. Such understanding is a critical tool in improving prevention efforts at the population scale [10].

A unique feature of the NVDRS, distinct from other mortality registries, is that most cases are accompanied by a textual "narrative" abstracted by NVDRS staff using original source documents including death scene investigations, interviews with people who knew the decedent, contents of suicide notes, autopsy reports, and related sources [8]. Each case in the registry has multiple narratives: one is primarily derived from coroner or medical examiner reports and a second is primarily derived from law enforcement investigations. These narratives thus provide qualitative textual evidence on a population scale. Previously, qualitative text data in suicide research was generally limited to small psychological autopsy studies [11] or interviews with people who had survived a suicide attempt [12]. However, a handful of studies have begun using these NVDRS text data, some leveraging analytic tools appropriate for

manipulating large amounts of text such as natural language processing (NLP) algorithms [13] but most applying traditional qualitative approaches (i.e., content analysis) to smaller subsets of the registry [14–18].

Regardless of the analytic approach used, any effort to draw inferences from the NVDRS narratives need to be made with a careful consideration of potential biases and limitations in data collection and measurement. From a data quality perspective, the NVDRS texts are unique, as they are explicitly written for research purposes by centrally-trained staff. NVDRS staff undergo regular training to enhance consistency of abstraction, and state data are reviewed centrally by CDC staff before they are made available to external investigators [19, 20]. However, these narratives may still be subject to measurement error which could bias inferences [21]. For example, if there are systematic patterns in the amount or quality of text written about each case as a function of decedent characteristics (e.g., age or race), this information bias would impact the validity of any conclusions drawn about how suicide mortality varies over the life course or how established risk factors for suicide (e.g., depression, substance misuse) relate to racial differences in suicide risk, respectively. Investigators need to understand the strengths and limitations of these narrative texts to appropriately account for any such sources of bias in their empirical research.

We aim to further scientific conversation about and harness the NVDRS's utility as a tool for informing suicide prevention efforts. Therefore, we investigated the relationship between decedent characteristics and length of NVDRS text narratives from nearly 240,000 suicide and undetermined deaths from 2003 to 2017. The length of the narrative is used to proxy the *information potential* of the text [22]. These findings can inform the work of investigators in their efforts to identify novel risk and protective factors for suicide.

## Methods

### Data source and elements

The NVDRS registry is publicly available through the CDC's Web-based Injury Statistics Query and Reporting System (WISQARS) [23]; however, the text narrative elements are only available to external investigators through a restricted-access data use agreement. We obtained NVDRS Restricted-Access Data (RAD) from the CDC in May 2020 using their application procedures [8, 10]. This dataset consisted of 239,716 deaths of all ages from suicide (including multiple suicides, and homicide followed by suicide), accidental firearm, and undetermined cause from 37 NVDRS sites (AK, AZ, CA, CO, CT, DE, DC, GA, GI, IL, IN, IA, KN, KY, ME, MD, MA, MI, MN, NV, NH, NJ, NM, NY, NC, OH, OK, OR, PA, RI, SC, UT, VT, VA, WA, WV, and WI), as well as Puerto Rico, from 2003 to 2017.

All data in the NVDRS registry, both quantitative variables and qualitative text narratives, are coded and written, respectively, by trained abstractors in each participating state [8, 10, 20, 24]. All data are generated using original source documents (death certificates, coroner or medical examiner reports, witness statements, law enforcement reports, scene investigations, etc.). These documents are converted into quantitative variables and qualitative text narratives using a common data entry system. The CDC provides centralized training for state abstractors, reviews the submitted data before it is released to external investigators, and has a set of quality assurance procedures to support reliable abstraction of documents across states and over time [19, 25].

**Qualitative text narratives.** This analysis used two types of narratives for each decedent: one primarily derived from coroner and medical examiner reports (C/ME), and one primarily derived from law enforcement investigations (LE). While these are both written by NVDRS staff and therefore should have similar information, we examined each type separately to assess

the degree to which any patterns we observe regarding decedent characteristics are similar across the two texts. If the patterns are similar, this may reflect features of the centralized NVDRS system or general limitations in the accuracy and completeness of mortality documentation (i.e., lack of access to specific records by NVDRS staff, incomplete death certificates) [26]. If the patterns differ, this may reflect characteristics of the source documents (e.g., toxicology reports, police reports) or reporting procedures. For example, not all decedents undergo autopsy, and states vary in whether they have local or centralized coroner and/or medical examiner systems, both of which would primarily influence the C/ME narratives. In addition, while the overwhelming majority of deaths are investigated by local, rather than state or federal, law enforcement agencies, most NVDRS sites do not have a pre-existing information-sharing infrastructure that would enable the seamless transfer of source documents between these police departments and the state NVDRS abstractors [25]. The net result is that NVDRS staff often must foster relationships with local stakeholders that create the source documents used for data abstraction (i.e., coroners, police departments) to ensure complete reporting. This may introduce systematic state and chronological differences in the completeness and length of the narratives as NVDRS staff foster and build these partnerships over time.

**Inclusion criteria.** Exploratory analyses confirmed that narratives for multiple deaths (i.e., multiple suicides, homicide followed by suicide) were longer than those of single deaths, and therefore these cases were excluded from analysis (n = 4,361). Because our analysis is focused on suicide, accidental firearm deaths were also excluded (n = 2,247). Undetermined cause deaths were retained in the analysis to reflect potential misclassification of suicide [27, 28]. As illustrated by Fig 1, after these exclusions the analytic sample size was n = 233,108, which consisted of single suicide deaths (n = 195,343) and undetermined deaths (n = 37,765). This was the sample used in *Analysis 1*, which examined predictors of whether the decedent was *missing a text narrative*.

*Analysis 2* examined predictors of the length (in characters, including spaces) of the narrative. For this analysis, the sample was additionally limited to those cases in which the NVDRS coders indicated that "circumstances were known," as the intent of the narrative is to provide a detailed description of these circumstances and the Data Users Guide specifies this condition

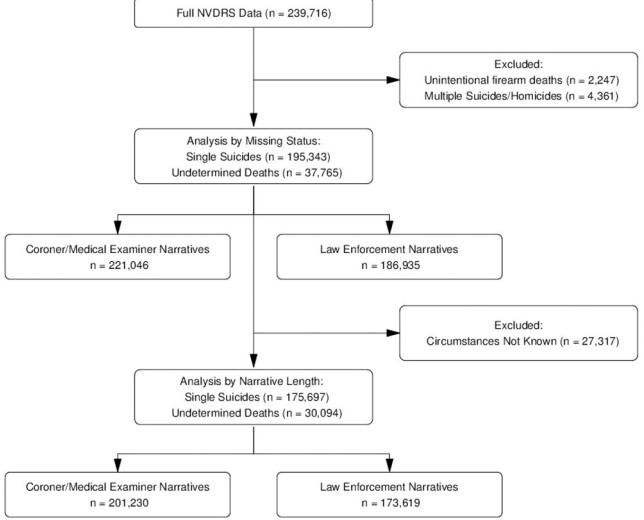

**Fig 1. Flowchart of sample inclusion/exclusion criteria for analyses of narrative texts, National Violent Death Reporting System, 2003–2017.**

should be applied. This resulted in the exclusion of an additional n = 27,317 cases. Through additional exploratory analyses we noted that there were several cases where the NVDRS data indicated that circumstances were "not known," but the case still had a narrative of at least 31 characters in length. A description and examples of these narratives are provided in the S1 Appendix. This means the results of the analysis of narrative length presented here are likely conservative. Also, in the S1 Appendix we provide a random sample of 10 annotated examples of short (31 to <200 characters) and long (>500 characters) narratives, to illustrate the notion that longer texts have more *information potential.*

This project was approved by the CDC-NVDRS, and this analysis was deemed exempt from human subjects regulation by the Institutional Review Board at the University of Michigan.

**Data access.** The narrative data used in this analysis are available by request from the CDC through their restricted-access data process. Other NVDRS data are publicly-available: https://www.cdc.gov/violenceprevention/datasources/nvdrs/datapublications.html. Cells with <5 observations have been suppressed in this publication, as required by the NVDRS Data Use Agreement.

## Predictors

The quantitative variables used in the regression analyses focused on seven decedent character-istics that are mandated on standard US Death Certificates [29]: age (coded as ≤18, 19–29, 30–39, 40–49, 50–59, 60–69, 70–79 and ≥80 years); sex (coded as female, male, or unknown); race/ethnicity (coded as American Indian/Alaskan Native, Asian/Pacific Islander, Black/Afri-can American, Hispanic, Non-Hispanic White, Two or more races, Other, or unknown); mari-tal status (coded as married/civil union/domestic partnership, separated/divorced, widowed, never married/single but not otherwise specified, or unknown); educational attainment (coded as 8th grade or less, 9th to 12th grade, high school diploma or GED, some college but no degree, associate's degree, bachelor's degree, master's degree, doctorate/professional degree, or unknown); military status (yes, no or unknown); and homeless (yes, no or unknown). In addition, the regression models adjusted for whether an autopsy was performed (yes, no, or unknown); location of death (home, hospital, hospice/nursing home, other, or unknown); and year (2017 as the reference). These additional variables were included because exploratory analyses indicated they improved both absolute and relative model fit. Because the amount of missing data in these predictor variables was generally limited (see Table 1), we included a dummy code for "missing" for all predictors so that these observations were retained in the regression analyses. The exception to this was for education level, which had substantial amounts of missingness; therefore, for this variable we conducted an additional analysis accounting for missing values using imputation with multivariate chain equations (30 datasets, 20 iterations). For all analyses, NVDRS site, which is an identification variable that reflects which state abstracted a particular case, was used as a clustering variable in the regression anal-yses, as described below.

## Analysis

We examined how the (i) percent of missing narratives and (ii) text character length among those with a non-missing narrative, for both C/ME and LE texts, varied as a function of dece-dent characteristics.

**Analysis 1: Predictors of missing narratives.** We conducted extensive exploratory analy-sis of the text narratives focused on the length of the C/ME and LE texts. While in most cases the narrative was simply missing (zero characters), in other cases the only text provided was

**Table 1. Decedent characteristics stratified by narrative missing status: Suicide and undetermined deaths in the National Violent Death Reporting System, 2003–2017.**

| | | C/ME Narratives | | LE Narratives | |
|---|---|---|---|---|---|
| | Total | Not Missing | Missing | Not Missing | Missing |
| | N = 233108 | N = 221046 | N = 12062 | N = 186935 | N = 46173 |
| **Age** (median [Q1; Q3]) | 45 [16; 83] | 45 [16; 83] | 47 [16; 84] | 45 [16; 83] | 46 [15; 84] |
| **Sex** | | | | | |
| Female | 57346 (24.6%) | 54629 (24.7%) | 2717 (22.5%) | 45051 (24.1%) | 12295 (26.6%) |
| Male | 175673 (75.4%) | 166381 (75.3%) | 9292 (77.0%) | 141860 (75.9%) | 33813 (73.2%) |
| Unknown/Missing | 89 (0.04%) | 36 (<0.1%) | 53 (0.4%) | 24 (<0.1%) | 65 (0.1%) |
| **Race/Ethnicity** | | | | | |
| White, non-Hispanic | 191214 (82.0%) | 181296 (82.0%) | 9918 (82.2%) | 154748 (82.8%) | 36466 (79.0%) |
| American Indian/Alaska Native | 3122 (1.34%) | 2855 (1.3%) | 267 (2.2%) | 2470 (1.3%) | 652 (1.4%) |
| Asian/Pacific Islander | 3855 (1.65%) | 3730 (1.7%) | 125 (1.0%) | 2966 (1.6%) | 889 (1.9%) |
| Black or African American | 18280 (7.84%) | 17528 (7.9%) | 752 (6.2%) | 13895 (7.4%) | 4385 (9.5%) |
| Hispanic | 12289 (5.27%) | 11786 (5.3%) | 503 (4.2%) | 9528 (5.1%) | 2761 (6.0%) |
| Other/Unspecified, non-Hispanic | 772 (0.33%) | 519 (0.2%) | 253 (2.1%) | 395 (0.2%) | 377 (0.8%) |
| Two or more races, non-Hispanic | 3268 (1.40%) | 3169 (1.4%) | 99 (0.8%) | 2830 (1.5%) | 438 (0.9%) |
| Unknown/Missing | 308 (0.13%) | 163 (0.1%) | 145 (1.2%) | 103 (0.1%) | 205 (0.4%) |
| **Education Level** | | | | | |
| 8th grade or less | 9515 (4.08%) | 8935 (4.0%) | 580 (4.8%) | 7259 (3.9%) | 2256 (4.9%) |
| 9-12th grade, no diploma | 24624 (10.6%) | 23266 (10.5%) | 1358 (11.3%) | 20248 (10.8%) | 4376 (9.5%) |
| HS or GED | 67738 (29.1%) | 64203 (29.0%) | 3535 (29.3%) | 55200 (29.5%) | 12538 (27.2%) |
| Some college, no degree | 26245 (11.3%) | 25069 (11.3%) | 1176 (9.7%) | 21634 (11.6%) | 4611 (10.0%) |
| Associate degree | 11708 (5.02%) | 11107 (5.0%) | 601 (5.0%) | 9563 (5.1%) | 2145 (4.6%) |
| Bachelor's degree | 17330 (7.43%) | 16647 (7.5%) | 683 (5.7%) | 14177 (7.6%) | 3153 (6.8%) |
| Master's degree | 5930 (2.54%) | 5668 (2.6%) | 262 (2.2%) | 4809 (2.6%) | 1121 (2.4%) |
| Professional or Doctorate degree | 2677 (1.15%) | 2577 (1.2%) | 100 (0.8%) | 2199 (1.2%) | 478 (1.0%) |
| Unknown/Missing | 67341 (28.9%) | 63574 (28.8%) | 3767 (31.2%) | 51846 (27.7%) | 15495 (33.6%) |
| **Marital Status** | | | | | |
| Married/In relationship | 74183 (31.8%) | 70101 (31.7%) | 4082 (33.8%) | 59275 (31.7%) | 14908 (32.3%) |
| Divorced/Separated | 55480 (23.8%) | 52718 (23.8%) | 2762 (22.9%) | 44955 (24.0%) | 10525 (22.8%) |
| Single/Never Married | 87033 (37.3%) | 83150 (37.6%) | 3883 (32.2%) | 70259 (37.6%) | 16774 (36.3%) |
| Widowed | 12832 (5.50%) | 12071 (5.5%) | 761 (6.3%) | 10041 (5.4%) | 2791 (6.0%) |
| Unknown/Missing | 3580 (1.54%) | 3006 (1.4%) | 574 (4.8%) | 2405 (1.3%) | 1175 (2.5%) |
| **Military** | | | | | |
| No | 177732 (76.2%) | 169255 (76.6%) | 8477 (70.3%) | 143528 (76.8%) | 34204 (74.1%) |
| Yes | 37527 (16.1%) | 35396 (16.0%) | 2131 (17.7%) | 30638 (16.4%) | 6889 (14.9%) |
| Unknown/Missing | 17849 (7.66%) | 16395 (7.4%) | 1454 (12.1%) | 12769 (6.8%) | 5080 (11.0%) |
| **Homeless** | | | | | |
| No | 218557 (93.8%) | 211183 (95.5%) | 7374 (61.1%) | 179221 (95.9%) | 39336 (85.2%) |
| Yes | 3083 (1.32%) | 3023 (1.4%) | 60 (0.5%) | 2601 (1.4%) | 482 (1.0%) |
| Unknown/Missing | 11468 (4.92%) | 6840 (3.1%) | 4628 (38.4%) | 5113 (2.7%) | 6355 (13.8%) |
| **Autopsy Performed** | | | | | |
| No | 97505 (41.8%) | 91110 (41.2%) | 6395 (53.0%) | 77869 (41.7%) | 19636 (42.5%) |
| Yes | 133969 (57.5%) | 128842 (58.3%) | 5127 (42.5%) | 108168 (57.9%) | 25801 (55.9%) |
| Unknown/Missing | 1634 (0.70%) | 1094 (0.5%) | 540 (4.5%) | 898 (0.5%) | 736 (1.6%) |
| **Place of Death** | | | | | |
| Home | 128517 (55.1%) | 122725 (55.5%) | 5792 (48.0%) | 107318 (57.4%) | 21199 (45.9%) |

*(Continued)*

**Table 1.** (Continued)

| | C/ME Narratives | | | LE Narratives | |
|---|---|---|---|---|---|
| | Total | Not Missing | Missing | Not Missing | Missing |
| | N = 233108 | N = 221046 | N = 12062 | N = 186935 | N = 46173 |
| Hospice or LTC | 2117 (0.91%) | 1898 (0.9%) | 219 (1.8%) | 893 (0.5%) | 1224 (2.7%) |
| Hospital | 40215 (17.3%) | 37988 (17.2%) | 2227 (18.5%) | 29165 (15.6%) | 11050 (23.9%) |
| Other | 60327 (25.9%) | 57511 (26.0%) | 2816 (23.3%) | 48691 (26.0%) | 11636 (25.2%) |
| Unknown/Missing | 1932 (0.83%) | 924 (0.4%) | 1008 (8.4%) | 868 (0.5%) | 1064 (2.3%) |
| **Circumstances known** | | | | | |
| No | 27317 (11.7%) | 19816 (9.0%) | 7501 (62.2%) | 13316 (7.1%) | 14001 (30.3%) |
| Yes | 205791 (88.3%) | 201230 (91.0%) | 4561 (37.8%) | 173619 (92.9%) | 32172 (69.7%) |

"Not available," "No report at this time," or "N/A" which are, in effect, missing values, as these texts were not describing salient characteristics that would be of interest to researchers. Therefore, we recoded all narratives with fewer than 31 characters (including spaces) to zero characters for analysis. After this recoding, there were 12,062 (5.2%) C/ME and 46,173 (19.8%) LE narratives treated as "missing" in the subsequent analysis; 6,170 observations (3%) were missing both C/ME and LE narratives. We then fit two logistic regression models (modeling C/ME and LE separately), to identify predictors of having a missing narrative (1 = missing, 0 = not missing), controlling for year, location of death, and autopsy status. There was significant clustering of the outcomes by site (intraclass correlation coefficient (ICC) for a missing narrative: C/ME = 0.57, LE = 0.48; ICC for narrative length: C/ME = 0.35, LE = 0.43). Therefore, we accounted for the clustering of observations within sites using Generalized Estimating Equations (GEE) modeling assuming an exchangeable correlation structure and a sandwich estimator to be robust against model misspecification [30]. GEE accounts for factors that cluster within sites (e.g., state demographic composition, C/ME system (centralized vs. local), abstracter experience). We also conducted a sensitivity analysis excluding sites with <5 observations missing a narrative (i.e., sites with nearly complete narrative data) to confirm that our analysis of missingness was not influenced by these sites.

**Analysis 2: Predictors of the length of the narratives among cases whose narrative was not missing.** The second analysis examined the predictors of the length of the C/ME and LE texts, as expressed by the count of characters (including spaces), conditional on having a non-missing narrative and having "known circumstances." The condition of "known circumstances" was applied as directed in the RAD Data User Guide and resulted in 27,317 cases excluded from this analysis (Fig 1). We used GEE quasi-Poisson models, with an exchangeable correlation structure and sandwich estimator, to examine the relationship between decedent characteristics and the length of the narratives while controlling for year, location of death, and autopsy status, separately for C/ME and LE narratives. The quasi-Poisson model is appropriate for outcomes that are discrete integers (i.e., count of character length) and are over-dispersed (i.e., variance greater than the mean) [31], as is the case in the present analysis. We also conducted a sensitivity analysis by excluding observations in the top 1% of character length (separately for C/ME and LE) to confirm that our analysis of length was not influenced by these outlier observations.

Finally, we conducted two additional post-hoc sensitivity analyses for both the missing narratives and narrative length to confirm that the robustness of our findings: (1) we additionally adjusted for presence of a toxicology report (coded yes vs. no/not applicable), which may result in longer narratives due to the description of substances, and (2) we re-ran all models excluding 30,094 undetermined deaths (that is, limiting the analysis to single-death suicide cases).

All analyses were conducted using R (version 4.0.2) and all p-values refer to two-tailed tests.

## Results

### Analysis 1: Predictors of missing narratives

Table 1 shows decedent characteristics of the overall analytic sample and stratified by whether their C/ME or LE narrative was missing. The sample was predominantly male and non-Hispanic white (NHW), with a median age of 46. Unsurprisingly, decedents whose characteristics were "unknown" were more likely to be missing narratives than those with valid data. However, even among decedents with known demographics there was variation in the number of missing narratives, although this variation was not always consistent across the two types of texts.

As shown by Fig 2 and S1 Table, after accounting for year, place of death, and autopsy status, there was a dose-response relationship between older age and relative odds of having a missing an LE, but not C/ME, narrative. Women were more likely to be missing LE (Odd ratio (OR): 1.12, 95% CI: 1.09–1.152), but not C/ME (OR: 1.01), narratives relative to men.

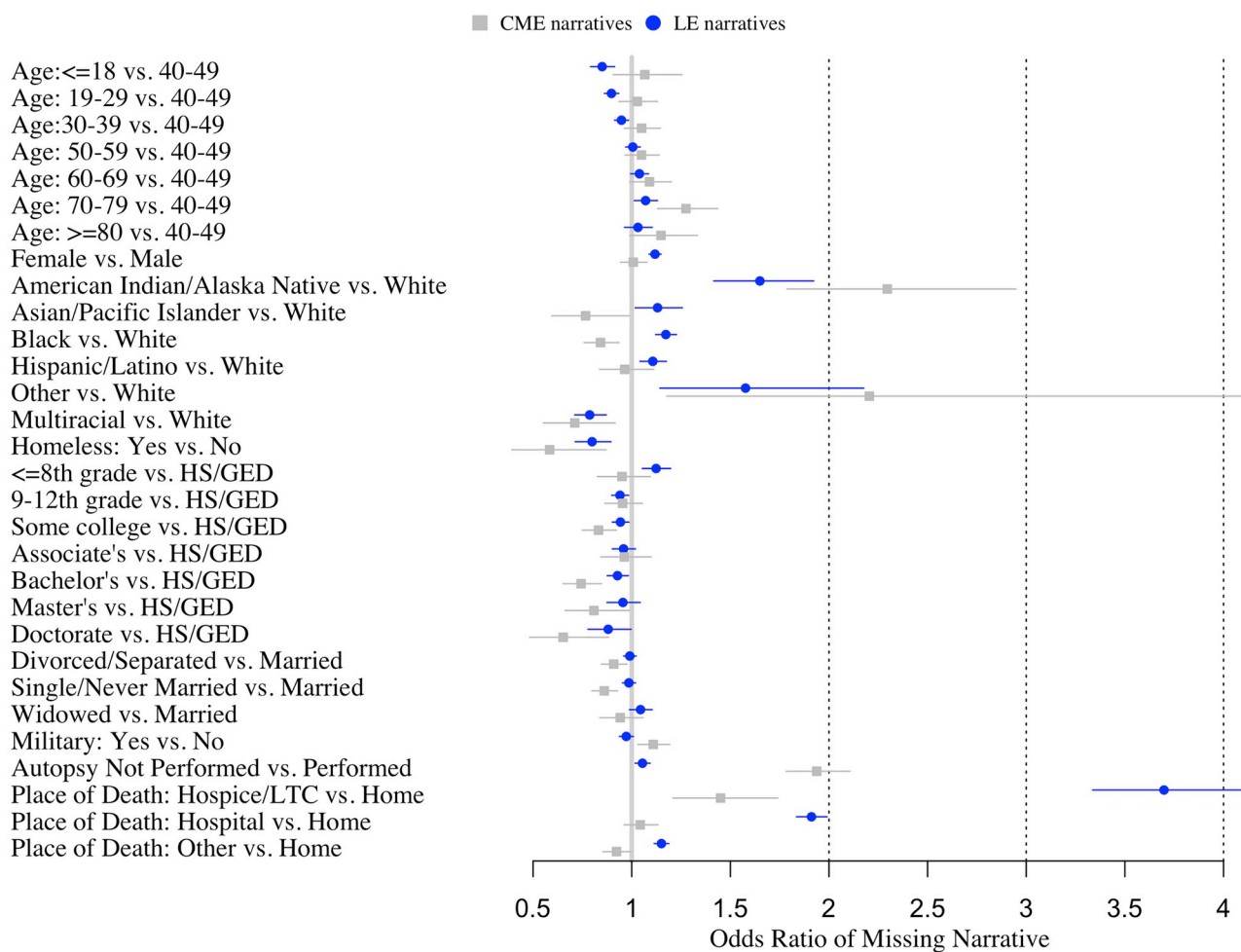

**Fig 2. Forest plot of relative odds (95% confidence intervals) of missing C/ME and LE narrative texts associated with decedent characteristics, NVDRS 2003–2017.** Estimates are adjusted for all variables show in the figure as well as year, location of death, and autopsy status and account for clustering within site using GEE with robust standard errors.

Decedents who were Native American/Alaskan Native were more likely to be missing both C/ME (OR: 2.30, 95% CI: 1.79–2.95) and LE (OR: 1.65, 95% CI: 1.42–1.92) narratives relative to NHW, while decedents who were Asian/Pacific Islander, Black, or Hispanic were more likely to be missing LE narratives but less likely to be missing C/ME narratives relative to NHW. Decedents with more education were consistently less likely to have missing narratives (e.g., $OR_{Doctorate\ vs.\ HS}$: 0.65, 95% CI: 0.48–0.88 for C/ME). Marital status and military history were not associated with missingness. As shown by S2 Table the results of the sensitivity analysis excluding sites with <5 observations missing a narrative (i.e., nearly complete narrative data) were consistent with the main results.

## Analysis 2: Predictors of the length of narratives

Table 2 shows decedent characteristics as a function of narrative count length, which for ease of interpretation is stratified into tertiles, among those with "known" circumstances.

Fig 3 and S3 Table show the results of the quasi-Poisson regression models, adjusted for site, year, place of death, and autopsy status. The estimates reflect the relative ratio (RR) of mean character counts. Older age was consistently associated with shorter narratives, as was being Black ($RR_{CME}$: 0.94, 95% CI: 0.93–0.95), or Asian/Pacific Islander ($RR_{CME}$: 0.97, 95% CI: 0.95–0.99) race relative to NHW and being single relative to being married ($RR_{CME}$: 0.98, 95% CI: 0.98–0.99). Females ($RR_{LE}$ = 1.05, 95% CI: 1.04–1.05) and those with more education had longer narratives (e.g., $RR_{Doctorate\ vs.\ HS}$: 1.05, 95% CI: 1.02–1.07 for C/ME). As shown by S4 Table the results of the sensitivity analysis excluding the longest outlier narratives were consistent with the main results.

S5–S8 Tables show the results of additional sensitivity analyses for missing CME and LE narratives (S5 and S6 Tables, respectively) and CME and LE narrative length (S7 and S8 Tables, respectively). Model 1 of these tables reprints our main analyses for ease of comparison. Model 2 shows estimates using imputed education level instead of dummy-coded missing status; the findings are largely unchanged using this imputed education variable, even if some point estimates are no longer statistically significant: higher education is inversely associated with the narrative being missing, particularly for the CME narratives, and, conditional on having a non-missing narrative, higher education is associated with longer texts for both CME and LE narratives. Model 3 provides the results from sensitivity analysis excluding all cases of undetermined cause of death and shows that findings were substantially unchanged from our main analysis. Finally, additionally adjusting for presence of a toxicology report (Model 4) had no substantive impact on our findings.

## Discussion

Decedent characteristics are significantly related to the presence and length of narrative texts for suicide and undetermined deaths in the NVDRS, even after accounting for variation across sites, length of time the site had been participating in this surveillance system, and characteristics of the death event (i.e., location of death, autopsy status). To our knowledge this is the first study to comprehensively examine how decedent characteristics relate to the quantity of narrative data in this registry. We found that even after accounting for differences across sites and post-mortem factors, decedents who were older, racial/ethnic minority, and had less education were more likely to have missing narrative texts. Further, even among those with a narrative, these characteristics were also predictive of shorter texts. These findings extend prior research in this registry that has examined how decedent characteristics relate to classification of cause of death (i.e., suicide vs. undetermined) [32] and factors that relate to the completeness of these data within specific states [33]. While this study cannot

**Table 2. Decedent characteristics stratified by narrative length: Suicide and undetermined deaths in the National Violent Death Reporting System, 2003–2017.**

| | C/ME Narratives (character length) | | | LE Narratives (character length) | | |
|---|---|---|---|---|---|---|
| | Short: 31–396 | Medium: 397–659 | Long: 660–9961 | Short: 31–402 | Medium: 403–731 | Long: 732–9985 |
| | *N = 67119* | *N = 67193* | *N = 66918* | *N = 57974* | *N = 57825* | *N = 57820* |
| **Age** (Median [Q1; Q3]) | 45.0 [17.0;83.0] | 46.0 [17.0;83.0] | 45.0 [16.0;82.0] | 46.0 [17.0;84.0] | 46.0 [17.0;83.0] | 44.0 [16.0;82.0] |
| **Sex** | | | | | | |
| Female | 14721 (21.9%) | 16672 (24.8%) | 18937 (28.3%) | 13716 (23.7%) | 13606 (23.5%) | 15040 (26.0%) |
| Male | 52398 (78.1%) | 50521 (75.2%) | 47980 (71.7%) | 44258 (76.3%) | 44219 (76.5%) | 42779 (74.0%) |
| Unknown/Missing | 0 (0.0%) | 0 (0.0%) | 1 (<0.1%) | 0 (0.0%) | 0 (0.0%) | 1 (<0.1%) |
| **Race/Ethnicity** | | | | | | |
| White | 55677 (83.0%) | 56071 (83.4%) | 55252 (82.6%) | 47565 (82.0%) | 48854 (84.5%) | 48614 (84.1%) |
| American Indian/Alaska Native | 482 (0.7%) | 782 (1.2%) | 1230 (1.8%) | 497 (0.9%) | 658 (1.1%) | 1042 (1.8%) |
| Asian/Pacific Islander | 997 (1.5%) | 1108 (1.6%) | 1234 (1.8%) | 911 (1.6%) | 815 (1.4%) | 1007 (1.7%) |
| Black or African American | 5780 (8.6%) | 4851 (7.2%) | 4023 (6.0%) | 5355 (9.2%) | 4027 (7.0%) | 2638 (4.6%) |
| Hispanic | 3059 (4.6%) | 3266 (4.9%) | 4141 (6.2%) | 2622 (4.5%) | 2515 (4.3%) | 3590 (6.2%) |
| Other/Unspecified, non-Hispanic | 112 (0.2%) | 130 (0.2%) | 144 (0.2%) | 120 (0.2%) | 86 (0.1%) | 107 (0.2%) |
| Two or more races, non-Hispanic | 970 (1.4%) | 966 (1.4%) | 877 (1.3%) | 876 (1.5%) | 857 (1.5%) | 809 (1.4%) |
| Unknown/Missing | 42 (0.1%) | 19 (<0.1%) | 17 (<0.1%) | 28 (<0.1%) | 13 (<0.1%) | 13 (<0.1%) |
| **Education Level** | | | | | | |
| 8th grade or less | 2449 (3.6%) | 1988 (3.0%) | 2134 (3.2%) | 2167 (3.7%) | 1755 (3.0%) | 1679 (2.9%) |
| 9-12th grade, no diploma | 6895 (10.3%) | 6577 (9.8%) | 7406 (11.1%) | 6407 (11.1%) | 6035 (10.4%) | 5910 (10.2%) |
| HS or GED | 16687 (24.9%) | 19852 (29.5%) | 22417 (33.5%) | 15448 (26.6%) | 17871 (30.9%) | 18372 (31.8%) |
| Some college, no degree | 5820 (8.7%) | 7643 (11.4%) | 9845 (14.7%) | 5713 (9.9%) | 6906 (11.9%) | 7861 (13.6%) |
| Associate degree | 2640 (3.9%) | 3148 (4.7%) | 4495 (6.7%) | 2268 (3.9%) | 2895 (5.0%) | 3861 (6.7%) |
| Bachelor's degree | 4076 (6.1%) | 5017 (7.5%) | 6426 (9.6%) | 3597 (6.2%) | 4415 (7.6%) | 5438 (9.4%) |
| Master's degree | 1336 (2.0%) | 1772 (2.6%) | 2215 (3.3%) | 1266 (2.2%) | 1448 (2.5%) | 1870 (3.2%) |
| Professional or Doctorate degree | 641 (1.0%) | 774 (1.2%) | 986 (1.5%) | 631 (1.1%) | 630 (1.1%) | 814 (1.4%) |
| Unknown/Missing | 26575 (39.6%) | 20422 (30.4%) | 10994 (16.4%) | 20477 (35.3%) | 15870 (27.4%) | 12015 (20.8%) |
| **Marital Status** | | | | | | |
| Married/In relationship | 22076 (32.9%) | 22254 (33.1%) | 20452 (30.6%) | 18628 (32.1%) | 18721 (32.4%) | 18436 (31.9%) |
| Divorced/Separated | 15828 (23.6%) | 16158 (24.0%) | 16989 (25.4%) | 14332 (24.7%) | 14009 (24.2%) | 14071 (24.3%) |
| Single/Never Married | 24418 (36.4%) | 24307 (36.2%) | 25347 (37.9%) | 20676 (35.7%) | 21226 (36.7%) | 22211 (38.4%) |
| Widowed | 3862 (5.8%) | 3765 (5.6%) | 3421 (5.1%) | 3527 (6.1%) | 3232 (5.6%) | 2618 (4.5%) |
| Unknown/Missing | 935 (1.4%) | 709 (1.1%) | 709 (1.1%) | 811 (1.4%) | 637 (1.1%) | 484 (0.8%) |
| **Military** | | | | | | |
| No | 48265 (71.9%) | 52402 (78.0%) | 54355 (81.2%) | 42370 (73.1%) | 44656 (77.2%) | 47069 (81.4%) |
| Yes | 11374 (16.9%) | 11044 (16.4%) | 10204 (15.2%) | 10056 (17.3%) | 9668 (16.7%) | 8993 (15.6%) |
| Unknown/Missing | 7480 (11.1%) | 3747 (5.6%) | 2359 (3.5%) | 5548 (9.6%) | 3501 (6.1%) | 1758 (3.0%) |
| **Homeless** | | | | | | |
| No | 64341 (95.9%) | 65109 (96.9%) | 64715 (96.7%) | 55499 (95.7%) | 56074 (97.0%) | 56286 (97.3%) |
| Yes | 710 (1.1%) | 835 (1.2%) | 1169 (1.7%) | 791 (1.4%) | 760 (1.3%) | 819 (1.4%) |
| Unknown/Missing | 2068 (3.1%) | 1249 (1.9%) | 1034 (1.5%) | 1684 (2.9%) | 991 (1.7%) | 715 (1.2%) |
| **Autopsy Performed** | | | | | | |
| No | 26061 (38.8%) | 29824 (44.4%) | 28333 (42.3%) | 22426 (38.7%) | 26367 (45.6%) | 24310 (42.0%) |
| Yes | 40682 (60.6%) | 37152 (55.3%) | 38382 (57.4%) | 35240 (60.8%) | 31232 (54.0%) | 33322 (57.6%) |
| Unknown/Missing | 376 (0.6%) | 217 (0.3%) | 203 (0.3%) | 308 (0.5%) | 226 (0.4%) | 188 (0.3%) |
| **Place of Death** | | | | | | |
| Home | 36826 (54.9%) | 38361 (57.1%) | 39718 (59.4%) | 32178 (55.5%) | 34270 (59.3%) | 35380 (61.2%) |
| Hospice or LTC | 830 (1.2%) | 407 (0.6%) | 384 (0.6%) | 340 (0.6%) | 247 (0.4%) | 213 (0.4%) |

*(Continued)*

**Table 2.** (Continued)

|  | C/ME Narratives (character length) | | | LE Narratives (character length) | | |
|---|---|---|---|---|---|---|
|  | **Short: 31–396** | **Medium: 397–659** | **Long: 660–9961** | **Short: 31–402** | **Medium: 403–731** | **Long: 732–9985** |
|  | *N = 67119* | *N = 67193* | *N = 66918* | *N = 57974* | *N = 57825* | *N = 57820* |
| Hospital | 11063 (16.5%) | 11632 (17.3%) | 10561 (15.8%) | 9476 (16.3%) | 8470 (14.6%) | 8298 (14.4%) |
| Other | 18067 (26.9%) | 16565 (24.7%) | 16128 (24.1%) | 15769 (27.2%) | 14560 (25.2%) | 13761 (23.8%) |
| Unknown/Missing | 333 (0.5%) | 228 (0.3%) | 127 (0.2%) | 211 (0.4%) | 278 (0.5%) | 168 (0.3%) |

determine why narrative length varies as a function of these characteristics, this variation has implications for studies that seek to leverage these data to understand salient factors for suicide risk both within and across groups.

This study also identified several system-level factors associated with the presence and length of the narratives which researchers should be aware of when using these texts to investigate suicide mortality. LE narratives were more likely to be missing than C/ME ones, and prior work has shown is more challenging for state NVDRS staff to collate reports from

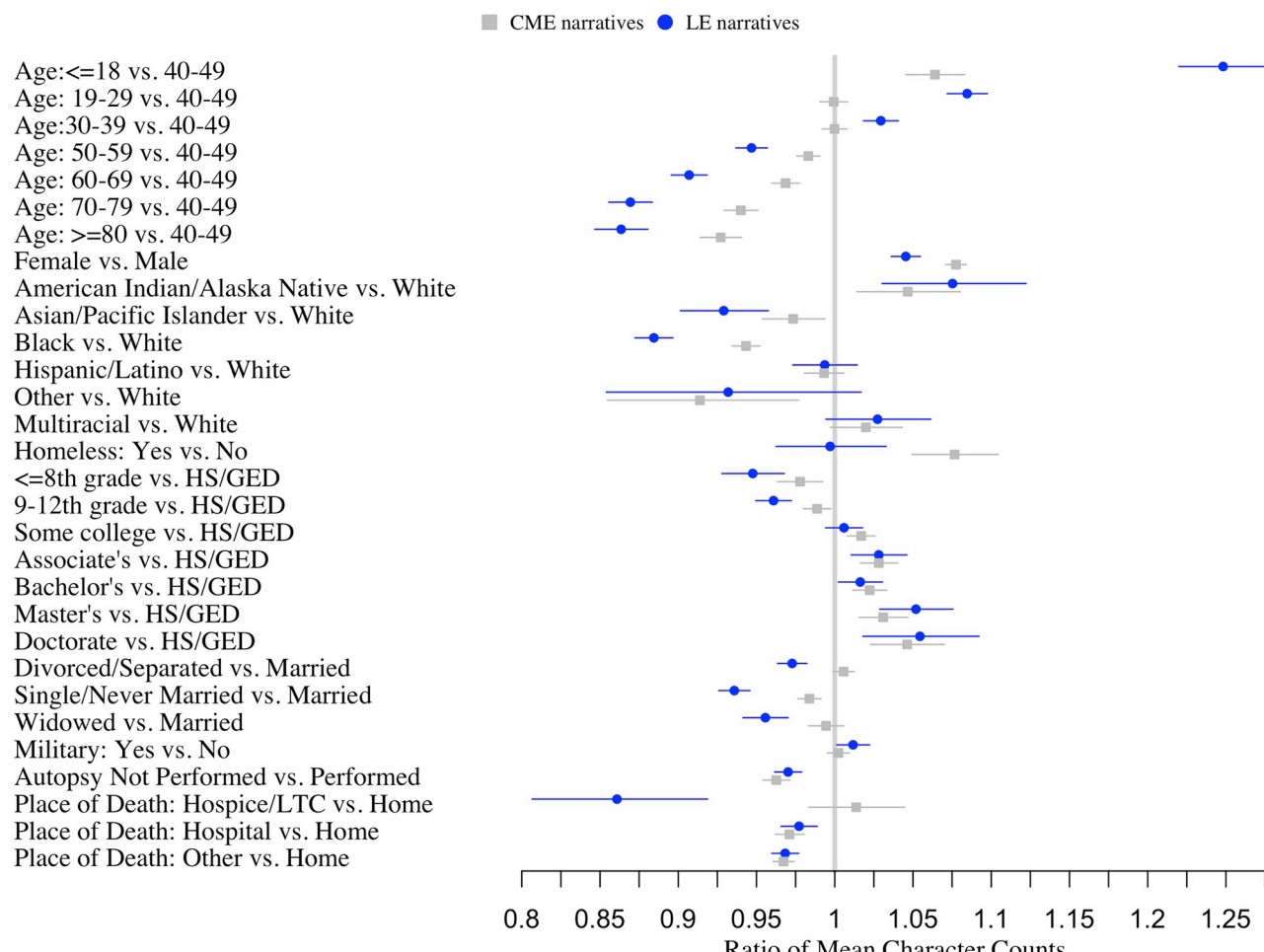

**Fig 3. Forest plot of relative ratios (95% confidence intervals) of the mean length of C/ME and LE narrative texts associated with decedent characteristics, NVDRS 2003–2017.** Estimates are adjusted for all variables show in the figure as well as year, location of death, and autopsy status and account for clustering within site using GEE with robust standard errors.

decentralized law enforcement systems [25, 33]. Conditional on having a narrative, C/ME narratives were substantially longer than LE texts, which may indicate they have more information potential for researchers seeking to identify novel risk factors. Sites that were newer to the NVDRS generated shorter narratives than those who had been in the system longer, potentially reflecting relative inexperience with writing these narratives or less established relationships with stakeholders (i.e., local law enforcement agencies) who provide the original source materials to the state NVDRS to abstract for the texts. Finally, while not part of the RAD that external researchers can access, there may be data processing variables that are created as part of the NVDRS abstraction process that internal staff could use to identify the specific reasons why a particular narrative is missing (e.g., indicators that the incident report needed follow-up; the specific document source; whether or not the document was available to the coder), which the CDC could use to identify system-level factors that contribute to data (in)completeness.

Suicide risk (attempts and mortality) has increased for the entire US over the past 20 years, particularly among Black adolescents [34] and middle-aged (age 45–64) adults [35]. Efforts to understand how these demographic characteristics intersect with known risk factors for suicidal behavior (i.e., depression, substance misuse, pain, loneliness, functional limitations, major life events), or, more importantly, to identify how these characteristics relate to modifiable protective factors, requires high-quality data at a population-scale. The NVDRS narratives are an important resource for researchers and policy makers as they seek to inform and implement evidence-based programs to reduce suicide risk, particularly to identify novel risk factors. For example, researchers have used the narrative texts to identify suicides related to transitioning into long-term care [13], intimate partner violence [15], risk factors among military personnel [17], and how multiple risk factors interact for middle-age men and women [14]. Such efforts are needed to address the stagnation in the field noted by Franklin et al. [6]. However, as this analysis indicates, there are systematic biases in the amount of information in these narratives as a function of decedent characteristics. Accounting for these biases will enhance the rigor of future studies that seek to extract the *information potential* of these narratives, whether using data science or traditional qualitative approaches.

Findings should be interpreted considering study limitations and strengths. First, this study cannot identify the reasons for the incompleteness or length of the narratives. For example, if police are less likely to be called to investigate the deaths of older decedents this could result in more missing or shorter LE narratives, but this cannot be determined from the registry data. Second, briefer narratives are not necessarily of poor quality; while it is beyond the scope of this analysis, future work should examine whether the information content in the narratives is related to decedent characteristics. This study also has several strengths. The large sample size and breadth of variables allowed us to explore variation across a wide range of decedent characteristics, and these findings can inform future data science (i.e., NLP) as well as traditional qualitative analysis of these narratives.

Although the NVDRS is a registry that is collated for researchers, the source documents it relies on to generate its data, both quantitative and qualitative (i.e., law enforcement reports, death certificates), were designed with a different purpose and are created by non-researchers (i.e., police officers, coroners, etc.). This is not a unique problem: for example, health services researchers routinely use insurance billing records to quantify the burden of disease and identify risk factors even though these records were designed for tracking healthcare payments. It is recognized that billing records have valuable information regarding population health and well-being, but also that these records are incomplete indicators of those constructs.

Conceptually, the NVDRS has complete catchment of suicide mortality in the United States. This potential makes it an invaluable resource for public health. However, the amount of information that is contained in this registry is uneven. Systematic patterns in incomplete

data, particularly across racial/ethnic groups, have been previously documented in mortality records [26, 36–38] and population health surveillance efforts (e.g., COVID infection and mortality [39]) The CDC and state NVDRS programs should examine why the information bias identified in this study occurs, and work with local, state, and federal stakeholders, as well as external researchers, to address it. Potential means of addressing the issues identified in this existing archive include the creation of sampling weights that account for differential selection (i.e., missingness) of having a narrative, and collaborating with data users to create trainings for researchers who want to use the narrative data to ensure their analytic approach minimizes potential biases. For future data abstraction in this archive, NVDRS sites should experiment with different approaches to incentive more complete data collection from local stakeholders and high-quality narrative abstraction. These text data have tremendous potential to provide new insights into suicide risk and minimizing information bias in will help ensure these narratives fulfill that potential.

## Supporting information

**S1 Appendix. Exploring the "information potential" of short and long narrative texts.**
(DOCX)

**S1 Table. Logistic regression of missing status for NVDRS narratives abstracted from coroner/medical examiner and law enforcement reports.**
(DOCX)

**S2 Table. Logistic regression of missing status for NVDRS narratives abstracted from coroner/medical examiner and law enforcement reports: Sensitivity analysis excluding sites with <5 missing narratives.**
(DOCX)

**S3 Table. Quasi-Poisson regression of character length of NVDRS narratives predicted by demographic characteristics.**
(DOCX)

**S4 Table. Quasi-Poisson regression of character length of NVDRS narratives predicted by demographic characteristics, excluding outliers (longest 1% of narratives).**
(DOCX)

**S5 Table. Sensitivity analyses for logistic regression of missing status for NVDRS narratives abstracted from coroner/medical examiner (CME) reports.**
(DOCX)

**S6 Table. Sensitivity analyses for logistic regression of missing status for NVDRS narratives abstracted from law enforcement (LE) reports.**
(DOCX)

**S7 Table. Sensitivity analyses of Quasi-Poisson regression of character length of NVDRS narratives abstracted from coroner/medical examiner (CME) reports.**
(DOCX)

**S8 Table. Sensitivity analyses of Quasi-Poisson regression of character length of NVDRS narratives abstracted from law enforcement (LE) reports.**
(DOCX)

## Acknowledgments

**Disclaimer:** The findings and conclusions of this study are those of the authors alone and do not necessarily represent the official position of the Centers for Disease Control and Prevention or of participating National Violent Death Reporting System (NVDRS) states. The NVDRS is administered by the Centers for Disease Control and Prevention by participating NVDRS states.

## Author Contributions

**Conceptualization:** Briana Mezuk, Viktoryia A. Kalesnikava, Tomohiro M. Ko, Cassady Collins.

**Formal analysis:** Viktoryia A. Kalesnikava, Jenni Kim.

**Funding acquisition:** Briana Mezuk.

**Methodology:** Jenni Kim.

**Project administration:** Briana Mezuk.

**Supervision:** Briana Mezuk.

**Visualization:** Viktoryia A. Kalesnikava.

**Writing – original draft:** Briana Mezuk.

**Writing – review & editing:** Briana Mezuk, Viktoryia A. Kalesnikava, Jenni Kim, Tomohiro M. Ko, Cassady Collins.

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
