## [Decision Letter · Decision Letter 0]

4 Feb 2021

PONE-D-21-00409

Not discussed: Inequalities in narrative text data for suicide deaths in the National Violent Death Reporting System

PLOS ONE

Dear Dr. Mezuk,

Thank you for submitting your manuscript to PLOS ONE. After careful consideration, we feel that it has merit but does not fully meet PLOS ONE’s publication criteria as it currently stands. Therefore, we invite you to submit a revised version of the manuscript that addresses the points raised during the review process.

Two expert reviewers with considerable research in the area have provided reviews of your paper.  They both see considerable merit in your paper but make suggestions for strengthening the analysis. One issue raised by both is a concern with state-level variation; both reviewers provide specific suggestions for further analysis.  Two additional critiques I would highlight would be the issue of the relationship of LE/CME missingness to missingness of other data (Reviewer 1) and the relationship of these findings to the larger issue of missing data in other official records (Reviewer 2).  Finally, a comment of my own -- it would be helpful to know the extent of overlap of the two types of missing (and nonmissing) data.  Please address all points raised by the reviewers.

We look forward to receiving your revised manuscript.

Kind regards,

Ellen L. Idler

Academic Editor

PLOS ONE

Journal Requirements:

Reviewers' comments:

Reviewer's Responses to Questions

**Comments to the Author**

1. Is the manuscript technically sound, and do the data support the conclusions?

Reviewer #1: Partly

Reviewer #2: Yes

2. Has the statistical analysis been performed appropriately and rigorously? 

Reviewer #1: No

Reviewer #2: Yes

3. Have the authors made all data underlying the findings in their manuscript fully available?

Reviewer #1: Yes

Reviewer #2: No

4. Is the manuscript presented in an intelligible fashion and written in standard English?

Reviewer #1: Yes

Reviewer #2: Yes

5. Review Comments to the Author

Reviewer #1: This paper is well-written, engaging to read, and addresses a basic (but important) question about the quality of NVDRS narrative data. With a few methodological changes, discussed below, I think that this paper contributes to the existing literature and may help current NVDRS abstractors address inequalities in data collection.

Before I continue, I want to preface this review by clarifying that my current knowledge of the NVDRS system is on the abstraction side--I work with a state NVDRS and SUDORS (an overdose-specific subset of the NVDRS) team to improve their data quality. This means that I have a good knowledge of the data itself, but less knowledge about the data version your team for analysis.

The first central methodological issue I see in this paper is a failure to fully address the state level variations in the data. Simply including a state dummy variable is not sufficient, as it does not address the fact that error terms will be clustered at the state level. While you acknowledge that there is state-level variation in the data, you need to build in either clustered error effects or potentially use a hierarchical model to fully address the variation. The NVDRS entry system is complex, and states have developed a wide variety of methodologies for submitting their data. Further, the difficulties of centralized versus local authority reporting means that some states may get electronic data downloads with CME reports (which often include written narratives), while others rely on scanned PDFs and fully manual abstraction. Finally, these reports are on tight schedules, so states that rely on fully manual abstraction may have less time to develop lengthy narratives. In short, I suspect state-level factors may be even more important than you suggested, and need a more robust inclusion in the model. If I were you, I would run a hierarchical model clustering at the state level, with the same outcomes/distributions.

My second methodological issue is more basic. It seemed odd to me that the missing narratives were not discussed in the context of other missing variables. LE narratives, in particular, were probably missing due to a lack of law enforcement investigation (which you do mention). If the data allows, restricting your analysis to only cases where some other data from the same source was entered would be helpful. Otherwise, it is not clear if your results are just picking up on a lack of investigation, rather than a specific issue with the narratives themselves. Your second analysis, which focuses on the length of narratives, helped address this problem, but I think you need to be a bit more specific in either addressing why you did not filter out cases without ANY LE information, or remove them. I suspect that there is a strong correlation--the NVDRS training strongly discourages abstractors from skipping the narrative if any LE or CME information exists. If that is true, your paper may need to more clearly acknowledge that the missing narrative problem is directly and solely driven by missing data problem. If the correlation is moderate, you could include other variable missingness (as a percent, maybe?) as a variable in your regressions.

Your work briefly acknowledged many of the issues I discuss above, and I think will be a strong and interesting paper once they are more squarely addressed. I would love to see a revision, and to share the final version with my team--I know they would find it interesting!

Reviewer #2: This paper uses NVDRS data to examine how decedent characteristics are related to the length of narratives contained in the data set. Increasing numbers of studies are using NVDRS narratives to shed light on circumstances surrounding suicide. Thus, this study, although primarily descriptive in nature, is useful in encouraging researchers to think through possible biases in analyses of narratives. In some sense, the findings are not terribly surprising – they are consistent with what we know to be true from undercounts in the Census and inaccuracies in other official sources of data. Those who are male and racial minorities are more likely to be excluded in both cases.

I have the following suggestions for improvement:

1. The authors use the length of the narrative (in terms of character count) as a proxy for the information potential and quality of text. There are limitations to this approach, as noted by the authors on page 25. It would be a useful addition to include a small random sample of narratives of different lengths to contextualize the differences in the quality of information contained in these narratives. Were any sensitivity analyses conducted to determine if the results differ if number of words (rather than number of characters including spaces) is used to proxy length?

2. P. 13: Given the possibility for coder bias, can the authors control for individual NVDRS coders and/or their length of experience? For example, if newly-added states to the NVDRS are more demographically diverse and less experienced coders are working on those narratives, it could skew results. I don’t know whether that’s the case but it’s one of several possibilities.

3. There are also important differences in the background of medical examiners and coroners which may affect the original reports. The study controls for states, thus capturing potential state differences but beyond state controls, have they considered other ways to capture geographic differences in death investigation systems. E.g. https://www.cdc.gov/phlp/publications/coroner/death.html. Data on county of residence of the decedent are included in the NVDRS.

4. The authors might consider putting Tables 1 and 2 in the appendix and the regression analyses currently in the appendix in the body of the paper. Tables 1 and 2 should indicate whether the differences (e.g. between non-missing/missing and between C/ME and LE) are statistically significant.

5. Regarding a dose response relationship between age and odds of a missing narrative “consistent for C/ME and LE texts” (p. 20, lines 244-246). According to the CI in Table 1, the effect of age on the C/ME missing is generally not significant, and there are no differences across the age groups in the effect. The patterns are different for LE missing.

6. As alluded to in #2, the controls are interesting in their own right. For example, there are significant differences across states and over time in the patterns of missingness and narrative length. At a minimum, it would be useful to provide more discussion and suggestions for future research as to why these differences exist. Some of the between-state difference may relate to the points raised on page 24 (some could be tested explicitly – e.g. time in system).

7. The discussion would also benefit from further explication of possible reasons as to why these patterns exist. For examples, studies of the Census undercount and/or inaccuracies in other official records would provide insight. The authors mention this only in the last sentence of the paper. It would be useful to synthesize and relate some of the possible explanations for these patterns in other sources to this analysis to provide a richer interpretation of the findings.

6. PLOS authors have the option to publish the peer review history of their article (what does this mean?). If published, this will include your full peer review and any attached files.

Reviewer #1: No

Reviewer #2: No

---

## [Author Response · Author response to Decision Letter 0]

20 Apr 2021

Comments from the editor

1. Two additional critiques I would highlight would be the issue of the relationship of LE/CME missingness to missingness of other data (Reviewer 1) and the relationship of these findings to the larger issue of missing data in other official records (Reviewer 2). Finally, a comment of my own -- it would be helpful to know the extent of overlap of the two types of missing (and nonmissing) data. 

Thank you for this comment. We woud like to clarify that our dataset includes three “types” of missing data: (1) Observations with a narrative character length of zero (C/ME, LE, or both) or whose circumstances were coded as “not known” by the NVDRS abtractors; (2) Observations whose narrative status was assigned as missing by the investigators based on having <31 characters, even though circumstances were coded as ‘known’ by the NVDRS abstractors; and (3) missing data on the predictors (age, sex, education, etc.).

To illustrate the extent of overlap in missingness for the first two types, have added the variable “Circumstances known” to the bottom of Table 1. It illustrates that of those observations whose C/ME narrative is coded as “missing,” 62.2% are missing because the circumstances were not known (first type of missing data) and 37.8% were coded as missing by the investigators because the narratives were <31 characters long (second type of missing data). These proportions are roughly reversed for the LE narratives. Overall, 6,170 (3%) observations were missing both C/ME and LE narratives (which we now state in the text - see Methods, Analysis 1). 

This third type of missing data (missing data on predictors) is addressed in our response to Reviewer #1, comment #2. 

The issue of missing data in mortality/administrative records more generally is addressed in our response to Reviewer #2, comments #6 and 7.

2. Please ensure that your manuscript meets PLOS ONE's style requirements, including those for file naming. The PLOS ONE style templates can be found at https://journals.plos.org/plosone/s/file?id=wjVg/PLOSOne_formatting_sample_main_body.pdf and https://journals.plos.org/plosone/s/file?id=ba62/PLOSOne_formatting_sample_title_authors_affiliations.pdf

We have made these style changes, as requested. 

In your revised cover letter, please address the following prompts: a) If there are ethical or legal restrictions on sharing a de-identified data set, please explain them in detail (e.g., data contain potentially identifying or sensitive patient information) and who has imposed them (e.g., an ethics committee). Please also provide contact information for a data access committee, ethics committee, or other institutional body to which data requests may be sent. Or b) If there are no restrictions, please upload the minimal anonymized data set necessary to replicate your study findings as either Supporting Information files or to a stable, public repository and provide us with the relevant URLs, DOIs, or accession numbers. Please see http://www.bmj.com/content/340/bmj.c181.long for guidelines on how to de-identify and prepare clinical data for publication. For a list of acceptable repositories, please see http://journals.plos.org/plosone/s/data-availability#loc-recommended-repositories.

We have responded to these prompts in the revised cover letter, as requested. 

 

Reviewer #1 Comments to the Author

This paper is well-written, engaging to read, and addresses a basic (but important) question about the quality of NVDRS narrative data. With a few methodological changes, discussed below, I think that this paper contributes to the existing literature and may help current NVDRS abstractors address inequalities in data collection. Before I continue, I want to preface this review by clarifying that my current knowledge of the NVDRS system is on the abstraction side--I work with a state NVDRS and SUDORS (an overdose-specific subset of the NVDRS) team to improve their data quality. This means that I have a good knowledge of the data itself, but less knowledge about the data version your team for analysis.

1. The first central methodological issue I see in this paper is a failure to fully address the state level variations in the data. Simply including a state dummy variable is not sufficient, as it does not address the fact that error terms will be clustered at the state level. While you acknowledge that there is state-level variation in the data, you need to build in either clustered error effects or potentially use a hierarchical model to fully address the variation. The NVDRS entry system is complex, and states have developed a wide variety of methodologies for submitting their data. Further, the difficulties of centralized versus local authority reporting means that some states may get electronic data downloads with CME reports (which often include written narratives), while others rely on scanned PDFs and fully manual abstraction. Finally, these reports are on tight schedules, so states that rely on fully manual abstraction may have less time to develop lengthy narratives. In short, I suspect state-level factors may be even more important than you suggested, and need a more robust inclusion in the model. If I were you, I would run a hierarchical model clustering at the state level, with the same outcomes/distributions.

Thank you for this comment. You are correct that there is significant clustering by state, which we quantified using the intraclass correlation coefficient (ICC), which quantifies the variation between vs. within states.

 ICC for C/ME missingness: 0.57, for LE missingness: 0.48

As a comparison, the ICC for clustering of C/ME missingness by incident year was only 0.01 and for LE missingness was only 0.03. This illustrates that the amount of missingness in narratives clusters within states, but not within years (time). As a result, incident year was included as a covariate in all models of narrative missingness, and state was used as a clustering variable. 

ICC for C/ME length: 0.35, for LE length: 0.43

As a comparison, the ICC for clustering of C/ME length by incident year was only 0.18 and for LE missingness was only 0.11. This illustrates that the length of narratives clusters within states, with some modest clustering within years (time). As a result, incident year was included as a covariate in all models of narrative length, and state was used as a clustering variable. 

In response, we we considered the following methods: 1) Quasipoisson model with robust standard errors, 2) multilevel modeling (in our case, Generalized Linear Mixed Model (GLMM)) and bootstrap while having state and year as random effects, and 3) a generalized estimating equation (GEE) with state as clusters, assuming exchangeable correlation structure and using sandwich estimator to be robustness of misspecification. All of the options are appropriate to address the violation of independence and homogeneity of variance. The first option works the best when the source of heteroscedasticity is unknown. However, in our case we know that the data came in clusters (state). 

Both the second option and third option are appropriate for our research question and data. GEE and GLMM in general give very similar results. The difference of the two is that GEE gives the population (here, state) averaged estimates of parameters and GLMM gives individual estimates (here, state specific) averaged estimates of parameters. Since we are not interested in estimating the state specific estimates of parameters, using GEE tends to be a common choice. GEE in general requires a large number of clusters ~40, and the NVDRS dataset has 37 states as clusters. 

Therefore, we refit our regression models using GEE logistic (for narrative missingness) and quasi-poisson (for narrative length, conditional on non-missingness) models with state as clusters, assuming exchangeable correlation structure and using sandwich estimator to be robustness of misspecification. 

We have updated the Methods and the text, tables and figures of the Results using these GEE models. We note that the point estimates are substantially unchanged from the first manuscript, but now the standard errors reflect the clustering of observations within states. Please see the revised manuscript.

Also please see our response to Reviewer #2, comment #6 in which we discuss the importance of local, state, and federal stakeholders to collaborate to identify, and address, the sources of these systematic biases in data completeness.

2. My second methodological issue is more basic. It seemed odd to me that the missing narratives were not discussed in the context of other missing variables. LE narratives, in particular, were probably missing due to a lack of law enforcement investigation (which you do mention). If the data allows, restricting your analysis to only cases where some other data from the same source was entered would be helpful. Otherwise, it is not clear if your results are just picking up on a lack of investigation, rather than a specific issue with the narratives themselves. Your second analysis, which focuses on the length of narratives, helped address this problem, but I think you need to be a bit more specific in either addressing why you did not filter out cases without ANY LE information, or remove them. I suspect that there is a strong correlation--the NVDRS training strongly discourages abstractors from skipping the narrative if any LE or CME information exists. If that is true, your paper may need to more clearly acknowledge that the missing narrative problem is directly and solely driven by missing data problem. If the correlation is moderate, you could include other variable missingness (as a percent, maybe?) as a variable in your regressions.

Thank you for this comment. Generally, there is reatlively little missing data on the predictors/covariates used in the analysis as they are largely drawn from death certificates, as shown in Table 1 (i.e., only 89 observations (0.04%) in the dataset in total (that is, not conditional on having a narrative) were missing the variable “sex”). In addition, we explicitly model any covariate missingness in our regression models (see Supplemental Tables 1 and 2) - that is, we do not exclude any cases because of missingness on covariates, but instead include a dummy-coded “missing” value for every predictor in our analysis.

Finally, the analysis of narrative length (quasi-Poisson modeling) uses a truncated distribution of narratives to examine length, wherein we removed any observation with missing narratives from this analysis (either truly missing (character length=0) or assigned by us as being missing (character length<31, a threshold we determined through exploring the content of texts across different lengths).

In response to this comment, we have clarified how we handled missing data on covariates in the Methods (reprinted below):

“While the amount of missing data in these predictor variables was generally small (see Table 1a), we included a dummy code for missingness for all predictors so that these observations were retained in the regression analyses.”

Reviewer #2 Comments to the Author

1. The authors use the length of the narrative (in terms of character count) as a proxy for the information potential and quality of text. There are limitations to this approach, as noted by the authors on page 25. It would be a useful addition to include a small random sample of narratives of different lengths to contextualize the differences in the quality of information contained in these narratives. Were any sensitivity analyses conducted to determine if the results differ if number of words (rather than number of characters including spaces) is used to proxy length?

In response to this comment, we have added annotated examples of a random sample of five short (31 to <200 characters) and five long (>500 characters) C/ME and LE narratives to the Appendix (which also provides examples of texts that have <31 characters). The annotation describes the elements and features of these texts and serves as a crude proxy of the “information potential” of the texts. 

We appreciate and understand the sentiment of this comment, but feel that a meaningful analysis of the information potential of these narrative texts is beyond the scope of this paper and would benefit from a data science (e.g., natural language processing, topic modeling) approach. We have added emphasis on this in the Discussion (please see revised text).

2. P. 13: Given the possibility for coder bias, can the authors control for individual NVDRS coders and/or their length of experience? For example, if newly-added states to the NVDRS are more demographically diverse and less experienced coders are working on those narratives, it could skew results. I don’t know whether that’s the case but it’s one of several possibilities.

Thank you for this comment. We do not have data on individual coders/NVDRS staff, unfortunately. However, our new analytic approach of accounting for clustering within states using Generalized Estimating Equations (GEE, see response to Reviewer #1) should account for factors like coder experience that cluster within site). Even with this new analytic approach, we still observe substantial disparities in narrative length by race/ethnicity in these data.

3. There are also important differences in the background of medical examiners and coroners which may affect the original reports. The study controls for states, thus capturing potential state differences but beyond state controls, have they considered other ways to capture geographic differences in death investigation systems. E.g. https://www.cdc.gov/phlp/publications/coroner/death.html. Data on county of residence of the decedent are included in the NVDRS.

Thank you for this comment. We believe our new modeling approach (GEE) is an appropriate means of addressing factors (like centralized vs. decentralized death examination systems) that vary across states. 

We also want to clarify that our intent in this analysis is to account for state (NVDRS site) clustering as an analytic issue, not to identify the reasons for that state clustering (which is a distinct research question). We want to characterize the NVDRS data archive as a whole as a means to inform future research. Therefore, we do not feel it is appropriate to examine sub-site factors (e.g., county of death) in this analysis.

Please also see our response to comment #6 below.

4. The authors might consider putting Tables 1 and 2 in the appendix and the regression analyses currently in the appendix in the body of the paper. Tables 1 and 2 should indicate whether the differences (e.g. between non-missing/missing and between C/ME and LE) are statistically significant.

The findings shown in Supplemental Tables 2 and 3 (regression tables) are identical to Figures 2 and 3 in the main text, and therefore we think including them in the main text would be redundant. In contrast, current Tables 1 and 2 provide a description of the narrative data in absolute terms (vs. the regression tables which only show relative differences) and therefore we feel they provide valuable information for the reader that is not present elsewhere in the Results. As such, we have not made this suggested change.

We have elected not to include p-values in Table 1 because the purpose of this table is to describe the sample, rather than to test any particular hypotheses, and it contains a lot of information already. The Supplemental Tables provide the 95% confidence intervals for all these comparisons while accounting for state clustering.

5. Regarding a dose response relationship between age and odds of a missing narrative “consistent for C/ME and LE texts” (p. 20, lines 244-246). According to the CI in Table 1, the effect of age on the C/ME missing is generally not significant, and there are no differences across the age groups in the effect. The patterns are different for LE missing.

Thank you for this comment. We have now corrected the text to reflect that the relationship between age and narrative missingness is limited to LE texts.

6. As alluded to in #2, the controls are interesting in their own right. For example, there are significant differences across states and over time in the patterns of missingness and narrative length. At a minimum, it would be useful to provide more discussion and suggestions for future research as to why these differences exist. Some of the between-state differences may relate to the points raised on page 24 (some could be tested explicitly – e.g. time in system).

While we agree with the sentiment of the comment, our goal is to draw attention to these patterns to external researchers, like ourselves, who are interested in addressing substantive scientific questions with this archive can do so in a manner that accounts for the information bias we have identified.

In response to this comment we have added the ICCs that show the amount of clustering by state (see response to Reviewer #1, comment #1) and added language to the Discussion regarding suggestions for future research and collaborations between data creators and data users (see revised text, last paragraph of the Discussion).

7. The discussion would also benefit from further explication of possible reasons as to why these patterns exist. For examples, studies of the Census undercount and/or inaccuracies in other official records would provide insight. The authors mention this only in the last sentence of the paper. It would be useful to synthesize and relate some of the possible explanations for these patterns in other sources to this analysis to provide a richer interpretation of the findings.

While we agree with the sentiment of the comment, our goal is to draw attention to these patterns to external researchers, like ourselves, who are interested in addressing substantive scientific questions with this archive can do so in a manner that accounts for the information bias we have identified. 

Moreover, we do not feel that our analysis can test the reasons for the patterns we observe, as we note in our discussion of study limitations, and that any comments we make as to their source would be speculative. That is, with the data we have we cannot capture important NVDRS system factors like updates to the data abstraction dashboard (i.e., introducing new variables, changing the coding of variables), changes in training, changes in staffing, new initiatives prioritizing data collection on certain groups, etc. that may contribute to these patterns. Other stakeholders (individual sites, the CDC) likely have data on these elements that likely could address these questions. Therefore, we feel collaboration with CDC and state NVDRS staff is the most effective means of identifying the reasons for these patterns - and addressing them in the future. 

In response to this comment, we have added additional text to the Discussion on the need for collaboration between data creators and data users in the NVDRS to maximize the scientific utility of this archive. Please see the final paragraph of the Discussion.

---

## [Decision Letter · Decision Letter 1]

26 May 2021

PONE-D-21-00409R1

Not discussed: Inequalities in narrative text data for suicide deaths in the National Violent Death Reporting System

PLOS ONE

Dear Dr. Mezuk,

Thank you for submitting your manuscript to PLOS ONE. After careful consideration, we feel that it has merit but does not fully meet PLOS ONE’s publication criteria as it currently stands. Therefore, we invite you to submit a revised version of the manuscript that addresses the points raised during the review process.

Both of the reviewers and I agree that most of the issues raised in the first round of reviews have been well-addressed, and that the manuscript is much improved.  Each reviewer however raises one or more remaining but minor issues that would further improve the paper.  Reviewer 1 would like to have a fuller disclosure of the data sources to which you did or did not have access -- this would be a very helpful step for future research in the area.  Reviewer 2 recommends addressing the missingness of the decedent characteristic of education, since it is higher than the other characteristics, and suggests a sensitivity analysis excluding deaths of undetermined cause.  Please either make these changes or explain why you are not doing so.

Overall, however the paper makes a strong contribution, and the Appendix with text examples is particularly enlightening.

We look forward to receiving your revised manuscript.

Kind regards,

Ellen L. Idler

Academic Editor

PLOS ONE

Journal Requirements:

Reviewers' comments:

Reviewer's Responses to Questions

**Comments to the Author**

1. If the authors have adequately addressed your comments raised in a previous round of review and you feel that this manuscript is now acceptable for publication, you may indicate that here to bypass the “Comments to the Author” section, enter your conflict of interest statement in the “Confidential to Editor” section, and submit your "Accept" recommendation.

Reviewer #1: (No Response)

Reviewer #2: (No Response)

2. Is the manuscript technically sound, and do the data support the conclusions?

Reviewer #1: Yes

Reviewer #2: Yes

3. Has the statistical analysis been performed appropriately and rigorously? 

Reviewer #1: Yes

Reviewer #2: Yes

4. Have the authors made all data underlying the findings in their manuscript fully available?

Reviewer #1: Yes

Reviewer #2: Yes

5. Is the manuscript presented in an intelligible fashion and written in standard English?

Reviewer #1: Yes

Reviewer #2: Yes

6. Review Comments to the Author

Reviewer #1: This version represents a substantial improvement over the original, and addresses all of my major concerns with the first version. I have only one lingering concern. While I understand and support the authors' decision to not speculate on the mechanisms behind narrative length, I still think it needs to be more explicitly stated, earlier than in the discussion, that narratives are generally missing because the underlying document/information needed to complete a narrative was missing--and that the presence of these documents can be ascertained in other ways. In my initial review, I assumed the authors had access to the LE/CME specific variables that the NVDRS collects (such as toxicology, death circumstances, etc), which would be an easy way to check the assumption that a lack of a narrative implies a lack of (access to) a CME/LE, but the writing suggests that those are not available to the authors. I thought they were available in the restricted-access version of the NVDRS data. They are correct that death certificates do not provide the needed information--in our state, even the "autopsy performed" variable does not necessarily reflect whether or not our team had access to an autopsy. If it is not possible within the scope of the authors' data access to adjust for the presence of CME/LE data, it should at least be noted that the data exists, and could be utilized by someone with different permissions or working with a different time period. If the authors did have access to these variables and chose not to use them, a more robust explanation of the rationale is needed. I think it would be appropriate to consider these variables even if they do not exist for portions of the study period (as NVDRS data collection does change frequently, as the authors noted). Some information on the correlation between missing source data and narrative length would be so helpful.

I think that these concerns could be addressed briefly in the text. The discussion surrounding state variation was great, and already touches on some of the reasons why abstractors may not have access to these documents, so it shouldn't take much tweaking to address the other variables. Either way, I think the narratives represent a better aggregate measure of data availability than any one variable--the discussion surrounding narrative length was particularly interesting. I also agree with Reviewer 2 and wish that the regression tables were available in the main text. I find them easier to interpret than the forest plots. However, that's more of a personal preference than a true problem with the paper.

Reviewer #2: The authors have done a good job of addressing concerns raised in the first set of reviews. I have just a couple of additional points of clarification:

1. Although the level of missingness for decedent characteristics other than the presence of a narrative is generally low, that is not the case for education (25-30% of cases lack this information). Given this high level of missingness and the fact that including a dummy variable to indicate missing status can lead to biased estimates (Paul Allison and others), I recommend that the authors use multiple imputation instead.

2. Did the authors conduct a sensitivity analysis to determine whether the substantive conclusions are changed if the deaths of undetermined cause are excluded? This would be a worthwhile check, and results could simply be reported in the text.

7. PLOS authors have the option to publish the peer review history of their article (what does this mean?). If published, this will include your full peer review and any attached files.

Reviewer #1: No

Reviewer #2: No

---

## [Editor Report · Decision Letter 2]

28 Jun 2021

Not discussed: Inequalities in narrative text data for suicide deaths in the National Violent Death Reporting System

PONE-D-21-00409R2

Dear Dr. Mezuk,

We’re pleased to inform you that your manuscript has been judged scientifically suitable for publication and will be formally accepted for publication once it meets all outstanding technical requirements.

Kind regards,

Ellen L. Idler

Academic Editor

PLOS ONE
---

## [Editor Report · Acceptance letter]

7 Jul 2021

PONE-D-21-00409R2 

Not discussed: Inequalities in narrative text data for suicide deaths in the National Violent Death Reporting System 

Dear Dr. Mezuk:

I'm pleased to inform you that your manuscript has been deemed suitable for publication in PLOS ONE. Congratulations! Your manuscript is now with our production department. 

Kind regards, 

on behalf of

Professor Ellen L. Idler 

Academic Editor

PLOS ONE